# Participant experiences of a low-energy total diet replacement programme: A descriptive qualitative study

**Nerys M. Astbury** [ID]*[⊙], **Charlotte Albury** [ID][⊙], **Rebecca Nourse**[¤], **Susan A. Jebb**

Nuffield Department of Primary Care Health Science, University of Oxford, Radcliffe Primary Care, Oxford, United Kingdom

⊙ These authors contributed equally to this work.
¤ Current address: Faculty of Health, School of Exercise & Nutritional Sciences, Deakin University, Melbourne, Australia
* nerys.astbury@phc.ox.ac.uk

## Abstract

### Introduction

The participants' experience of low-energy total diet replacement (TDR) programmes delivered by lay counsellors in the community for the routine treatment of obesity is currently unclear. We interviewed a sample of twelve participants who took part in the Doctor Referral of Overweight People to Low-Energy total diet replacement Treatment (DROPLET) trial and were randomised to the TDR programme.

### Methods

We purposively sampled twelve patients who took part in the DROPLET trial, and conducted in-depth telephone interviews, which were audio-recorded and transcribed verbatim. Interview questions focused on participants' experiences and perceptions of the TDR programme. We conducted a thematic analysis, actively developing themes from the data, and used the one sheet of paper (OSOP) technique to develop higher-level concepts.

### Results

Nine key themes were identified; Reasons for taking part, Expectations, Support and guidance from the counsellor, Time to build a personal relationship, Following the TDR Programme, Adverse events, Outcomes from the TDR, Weight Loss Maintenance, Recommending TDR to others. The relationship between participants and the counsellor was central to many of the themes. Close relationships with counsellors facilitated TDR adherence through providing one-to-one support (including during difficult times), sharing expert knowledge, and building a close relationship. Adherence was also supported by the rapid weight loss that patients reported experiencing. Overall participants reported positive experiences of the TDR, and emphasised the positive impact on their wellbeing.

**Data Availability Statement:** The datasets used in the current study cannot be shared publicly because they contain potentially sensitive and identifiable patient information. Data are available on request from the University of Oxford, Nuffield

Department of Primary Care Data Access Committee for researchers who meet the criteria for access to confidential data (information. guardian@phc.ox.ac.uk).

**Funding:** NMA and SAJ are supported by funding from the NIHR Oxford Biomedical Research Centre (BRC). RN and SAJ were supported by funding from NIHR Applied Research Care (ARC) Oxford. SAJ is a NIHR) senior investigator. The main DROPLET trial was funded by a research grant from Cambridge Weight Plan Ltd UK to University of Oxford. We conducted this sub-study independently from the main trial, with separate funding from National Institute for Health Research (NIHR) Applied Research Care (ARC) Oxford. The funders had no input into the design, data collection, analysis decision to publish or preparation of the manuscripts. The authors had a right to publish regardless of the results. The views expressed are those of the author(s) and not necessarily those of the NHS, the NIHR or the Department of Health and Social Care.

**Competing interests:** SAJ was Chief Investigator on the DROPLET trial, funded by a research grant from Cambridge Weight Plan UK Ltd to the University of Oxford. She is an investigator on publicly-funded research studies in which a weight-loss intervention has been provided by WeightWatchers, Slimming World and Rosemary Conley free of charge. She attended a one-day meeting on digital health interventions, organised by Oviva, in which expenses were paid to the University of Oxford. She is a member of the NHS England advisory board relating to weight-loss interventions for the prevention and treatment of type 2 diabetes. NMA was co-investigator on the DROPLET trial funded by a research grant from Cambridge Weight Plan Uk Ltd to the University of Oxford. She is also co-investigator on a publicly funded research study in which weight-loss interventions have been provided by Slimming World and Rosemary Conley free of charge. RN and CA have no conflicts to declare. NMA was co-investigator on the DROPLET trial funded by a research grant from Cambridge Weight Plan Uk Ltd to the University of Oxford. She is also co-investigator on a publically funded research study in which weight-loss interventions have been provided by Slimming World and Rosemary Conely free of charge. CA and RA declare they have no competing interests.

## Discussion

Patients reported that a TDR programme delivered by lay counsellors in the community was a positive experience and effective in helping them to lose weight. Future trials should consider the central role of the person providing support and advice as a key component in the programme.

## Introduction

Recent clinical trials have demonstrated that significant and substantial weight loss can be achieved using a combination of a low-energy total diet replacement (TDR) diet and behavioural support programme [1–3]. Further studies have shown that this type of approach is a clinically and cost effective option for the treatment of obesity [4]. However, in many countries, clinical guidelines do not currently recommend that this approach should be used as a routine treatment for the management of obesity. Little is known about the experience of patients who are offered this treatment as part of routine care.

The aim of the Doctor Referral of Overweight People to Low Energy total diet replacement Treatment (DROPLET) trial was to determine clinical effectiveness of TDR programme for obesity in primary care. Doctors identified people who may benefit from weight loss. If they decided to join the trial and were randomised to the TDR intervention they were provided with the contact details of a local lay counsellor whom they should contact to arrange a in a series of one-to one appointments. A total of eight counsellors were involved in delivering the DROPLET intervention. For convenience, participants were assigned a counsellor in their local area and counsellors were paid for delivering the intervention by the company providing the intervention (Cambridge Weight Plan UK Ltd).

During the appointments, the counsellor provided participants with the formula products used to replace meals as well as providing behavioural support. Counsellors were not medically qualified (e.g. registered nurses or physicians), but had personal experience of successfully losing weight using a TDR programme, and were recruited on this basis. They had undertaken training on how to deliver a behavioural support programme and were all experienced in providing a similar TDR weight loss intervention as part of an established business. Some participants had prior knowledge of the brand.

The results of the DROPLET study demonstrated that after one year weight loss in participants randomised to the TDR group was 10.7kg (95% CI) compared with 3.1 (95% CI) in those randomised to receive usual care (weight-loss support from a nurse at their primary care practice). Furthermore an economic analysis found the TDR programme met the NICE standards for cost effectiveness typically used by the NHS in the UK [4] opening up the possibility of adopting this treatment into routine provision for weight loss.

Here we explore the thoughts, feelings and experiences of a sample of participants who were offered a TDR weight loss programme as part of the DROPPLET trial. In particular, we sought to understand more about their opinions and experiences of the TDR intervention which was delivered by lay counsellors in partnership with primary care teams who conduct the initial referral.

## Methods

The DROPLET study was a pragmatic randomised controlled trial conducted in 10 GP practices in Oxfordshire, UK. The aim of the study was to determine the clinical effectiveness,

feasibility and acceptability of referral to a commercial low-energy TDR programme compared with usual weight management interventions in primary care. The study was reviewed and approved by Health Research Authority (HRA) NHS Research Ethics Committee South Central (REF: SC/15/0337) and was prospectively registered on ISTCRN registry (ISRCTN 75092026.). All participants provided written consent before they were enrolled on the trial. To minimise participant burden all interviews took place over the telephone so it was not possible to obtain written consent from the participants who took part. The ethics committee approved the use of oral consent. The telephone conversation recording consent was transcribed verbatim to document the process.

## Total diet replacement programme

In response to a letter from a GP, participants were enrolled on to the trial by a practice nurse, before being individually randomised to a TDR programme delivered by a commercial provider (Cambridge Weight plan UK Ltd) or usual care (UC).

Participants allocated to the TDR group were referred to a local lay counsellor who invited the participant to attend regular appointments for 24 weeks. Further detail on the programme structure is reported in the study protocol [5]. For the first 12 weeks, participants met with the counsellor weekly for support, which comprised goal setting, feedback, encouragement, reassurance, and problem solving. During the TDR phase in weeks 1–8, participants replaced all food with formula food products supplied by the counsellors. During weeks 9–13 counsellors advised participants on how to safely reintroduce conventional food and reduce reliance on the formula products in a four week food reintroduction phase. During the weight maintenance phase from week 13 to 24, counsellors encouraged participants to attend monthly appointments and to consume one formula food product a day, with advice on how to maintain weight following weight loss. This programme was provided free of charge to participant up to week 24.

## Sampling strategy and participants

All participants in the main DROPLET trial were invited to take part by letter from their GP. All participants had a BMI >30kg/m$^2$, with patients who were taking insulin excluded from participation. Full details of the inclusion/exclusion criteria have been previously reported in full elsewhere [5]. During the consent process for the DROPLET trial, participants had the opportunity to opt-out of being approached by the researcher to take part in this qualitative study. None of the participants opted-out of being approached to take part in this qualitative study. We invited a purposive sample of participants via letter to take part in this qualitative study. We aimed to obtain a sample with similar baseline demographics to the participants in the main DROPLET trial. A total of twelve participants were approached to take part in this qualitative study. None of the people approached to take part in the interviews refused, and none of the participants who agreed to take part dropped out. Participants were approached and interviews conducted in batches of three. The intent was to recruit and interview sufficient numbers of participants to reach data saturation. Regulatory approvals were in place to interview up to 30 participants, but data saturation was reached after 10 participants, and we interviewed an additional 2 participants to ensure no new concepts were identified, which confirmed saturation.

The main study commenced in January 2015, with data-collection for the present study was initiated in June 2015. Participants were told that the purpose of the study was to explore their thoughts and feelings of the experience of taking part in the DROPLET study, regardless of whether these were good or bad.

## Interviews

The female trial manager (NMA) with PhD in Biomedical Sciences (Nutrition) and experience of working in clinical trials related to weight management and a female research assistant (RN), who had master's degree in Public Health Nutrition and prior experience of behavioural interventions conducted the semi-structured interviews with participants. Neither had been in personal contact with the participants. Participants only contact with the interviewers prior to the interviews was as part of the recruitment to the qualitative study and not in any part of delivering the intervention within the trial. Participants were explicitly told that the interviewers had no affiliation with the treatment provider or counsellors, nor with the participants own GP or GP practice. The interviewers both had training and experience in qualitative interviewing, and knowledge of the DROPLET trial. All interviews were conducted over the phone and audio-recorded and field notes were taken after each interview. Interviews were transcribed verbatim. They were not shared with participants to reduce participant burden as they were also taking part in a large trial with significant questionnaire burden.

Interviews took between twenty-eight and forty-six minutes, and focussed on exploring participants' expectations and experiences of the TDR programme. We developed a semi-structured interview topic guide based on the experience of the team, and the wider qualitative literature on weight management (S1 Material). The interview guide was piloted with colleagues prior to its use with participants. The schedule consisted of twelve open-ended questions with prompts to explore participants' experience of participating in the DROPLET trial and in particular, views on the TDR programme and type of weight-loss support. The interviewers discussed data together throughout the process, and reviewed transcripts as they were returned. Saturation occurred after ten interviews, and a further two were conducted to ensure no new concepts were identified, which confirmed saturation.

## Analytic approach

The analytical strategy aimed to understand people's subjective views on barriers and facilitators to undertaking a TDR weight loss programme referred by a GP, and delivered in the community by trained lay counsellors. The end goal was to understand the acceptability of the programme for participants in the DROPLET trial [6–8].

We followed a qualitative descriptive approach to analysis. Our ontological position was relativism, and our epistemological assumptions were grounded in subjectivism. In following this positioning, we viewed this research as both inductive and subjective. We remained aware of the active role of the researchers in co-creating data as it was generated, and also in influencing interpretation through the process of analysis.

In qualitative descriptive analysis, the researcher seeks to identify and learn about how people experience an event, or a process, or to learn about people's perspectives, rather than to generate theory [9]. This method offers opportunity to learn about and to describe phenomena about which little is known [8]. This was appropriate for our research as we do not know about participant experiences of this TDR programme delivered within a routine care context. We did not intent to generate new theories or concepts, but rather to identify how participants in the DROPLET trial coped with the TDR, including what barriers and facilitators they experienced to adherence. As part of this process the researchers aim to stay as close as possible to the "surface of the data and events" so that description of experiences also remain close to participant reported experience [7].

We followed the process of thematic analysis [10] to code, categorise, and identify and describe patterns in our data, but in line with our descriptive approach, we did not draw on or generate theory. The interviews were analysed by a female social scientist (CA) with discussion

throughout the process from NMA. CA is experienced in qualitative research which focusses on weight loss interventions and has a doctorate in primary health care. CA and NMA listed to all interviews and read interview transcripts. CA conducted line-by-line coding of each transcript, iteratively developing a data driven coding frame. CA and NMA grouped codes into broader categories, and then used the One Sheet of Paper (OSOP) technique [11] to identify key issues reported during interviews. We used a number of techniques to enhance trustworthiness and credibility of analysis. This included a coding record that was kept by CA to record how and why new codes were created, and initial thoughts when reading each transcript [12]. This record was shared with NA for peer-discussion of coding decisions [13]. Secondly, CA kept a reflexivity log, recording personal beliefs, assumptions, and experiences in order to be cognizant of these during analysis, and raising these with NMA during peer-discussions [14]. Finally thematic grouping were discussed with all authors to confirm their agree. Data were coded and managed using Nvivo V11. To assist anonymization, pseudonyms have been provided for personal names, brand names in all quotations presented here.

## Results

Twelve participants were interviewed (3 Male, 9 Female), all were White British and middle aged (range 40–75 years). To maintain the anonymity of the participants we have used pseudonyms to identify quotes.

All eight of the counsellors who were involved in delivering the TDR programme are represented in the sample. Three participants had a pre-existing diagnosis of type 2 diabetes and two had hypertension at the time they enrolled on the trial. At 1 year the average weight loss in these participants was -15.4 kg (10.3) which is somewhat greater than the average for the whole sample, and ranged from weight loss of 32.3kg to a weight gain of 1.9kg (Table 1).

The thematic analysis of audio recordings of the interviews identified nine key themes.

### Reasons for taking part

Participants reported the main reason for taking part was that they wanted to lose weight. This was either because they did not like the way that they looked, thought they would look better if

**Table 1. Characteristics of the interviewed participants.**

| Pseudonym | Gender | Ethnicity | Age (years) | Baseline BMI (kg/m$^2$) |
|---|---|---|---|---|
| Margo | F | White British | 65 | 42.1 |
| Christine | F | White British | 63 | 38.6 |
| Bernard | M | White British | 58 | 47.8 |
| Diane | F | White British | 63 | 32.1 |
| Felicity | F | White British | 61 | 34.3 |
| Jasmine | F | White British | 37 | 32.8 |
| Jack | M | White British | 51 | 33.3 |
| Georgia | F | White British | 45 | 36.4 |
| Natalie | F | White British | 40 | 38.8 |
| Simon | M | White British | 75 | 33.5 |
| Beth | F | White British | 62 | 40.8 |
| Jess | F | White British | 53 | 40.9 |
| | | Mean (SD) | 56.1 (11.2) | 37.3 (4.9) |

they lost weight, or because they were aware that they had health conditions which could be improved with weight loss.

> *Felicity: It was, basically I didn't like the way I looked anymore, um, I put on so much weight, um, and, when I went to see the doctors and we started um, talking about, um, an issue I had um, she just said to me would I be interested, I jumped at the chance.*

> *Simon: Well, the—I had—Since I was diagnosed with, with the diabetes problem, I knew I had to lose some weight.*

Participants desire to lose weight was not initiated by the trial. Many people described that they had wanted to lose weight for a long time, and had made previous, sometimes repeated attempts at weight loss that had not been successful, or only successful in the short-term. They responded to the invitation letter to take part in the trial because they thought it was an opportunity to get some help to lose weight from the health system.

> *Bernard: I'd tried other diets and other methods to lose weight, but I always put the weight back on or, I would fail within sort of a short period of time.*

Some people described that this particular attempt at weight loss was motivated by GP-endorsement. One participant reported being overweight for a long time, had tried losing weight in the past, but had resigned to not planning on another attempt. However, concerns expressed by the GP, along with the recommendation to consider the DROPLET trial encouraged them to take part.

> *Diane: Because I've been overweight for a long time. Up and down like a flipping yo-yo. And I was recommended the study by my GP, who obviously was concerned that I was overweight, as well. And that's really what kind of spurred me on a bit more this time.*

During interviews, participants were asked about their thoughts on the research component of the study. As reported in other studies, reasons for wanting to take part were often linked not only to personal benefit, but also to the consideration that through participation in a trial, they could be part of something bigger which could help other people [15, 16].

> *Felicity: I know it's quite a big study, and if I'm part of that study, and it might help somebody else in the future.*

> *Jasmine: Feeling like I'm doing it for science I felt like it was worthwhile, and bigger than just, you know, trying to be a bit slimmer.*

## Expectations

Participants expected large weight losses. Almost all participants reported positive feelings upon being randomised to the TDR, stating that they were 'overjoyed' or 'over the moon' on hearing the allocation. The reason many participants attributed to these positive feelings was that they wanted to lose a lot of weight, and they expected that the TDR programme was more likely to achieve this than the alternative usual care programme led by the practice nurse or other diets they had tried, or heard about. This was either because they had previous knowledge or experience of TDR, or that they knew a friend or family member who had lost a large amount of weight following a similar programme.

*Bernard: I was—I was overjoyed, because that's what I wanted. I wanted something that would be severe. And drastic. And I'd read—Once I'd heard about the programme, I read up a lot about it, and researched it. So I knew it was quite—it was quite drastic and severe. So yeah, that's what—I was quite, really happy that I got that.*

*Interviewer: And uh, which of the treatment groups were you allocated?*

*Felicity: I was actually allocated the [TDR], which, was amazing because that was the one I really, really wanted to do, so I couldn't believe it when I was actually chosen to do that one.*

*Interviewer: Okay, so, could you tell me a little bit more about how you felt, when the TDR er, diet, was given to you?*

*Felicity: Ecstatic I think. To be honest, I had a colleague of mine who had also, gone to do the study and was put on the [TDR] diet and, it was a gentleman, and I couldn't believe the results that he was seeing*

Only one participant reported that they would have preferred to be allocated to the usual care programme, delivered by the practice nurse rather than the TDR programme. However, like the other participants, they reported the desire to lose large amounts of weight. They had previously tried a TDR 'years and years ago' which involved a very severe energy restriction to (<300 calories a day). They would found this to be 'hugely unhealthy' and very expensive, and could not stick to it. They also reported a good relationship with their nurse, and reluctance to see someone new. As a result, they were hesitant about being allocated to the TDR in the DROPLET trial, and described hoping to be randomised to the usual care group.

*Diane: Initially, I was very disappointed. Because I was hoping to be in regular contact with the practice nurse about this. And obviously as I was on the (TDR) plan, as opposed to following a—I don't know what you call it, but yeah I suppose like a regular diet, but it was monitored by the practice nurse. So therefore I wouldn't see her very often, and it meant that I was dealing with somebody new. And I was quite hesitant over that.*

## Support and guidance from the counsellor

All participants randomised to the TDR group received one-to-one support form a trained TDR lay counsellor, who had had their own experience of losing weight on the TDR. When talking about their experience of the programme, all participants highlighted that the counsellor was the most important aspect of taking part. Participants' relationship with the counsellor was an important part of their success and positive experience of the TDR programme.

Participant's initial contact with the counsellor was on the phone to arrange their first session. After this, most people met their counsellor in person for at least half an hour once a week, at a time and location of their choosing. The few people that reported feeling initially apprehensive about starting the TDR programme said they felt more confident and ready to begin the programme after their first conversation with their counsellor. Participants highlighted the in-depth knowledge that the counsellor had about the programme, and felt they had confidence in their individual counsellors to help them to succeed:

*Jess: They were, brilliant, actually. They were very, are a lovely person. And they were very knowledgeable about their, you know, the products and things. So I got a great deal of confidence from them, to be honest.*

*Bernard: The only thing I will say, is—it was—it turned out better (than expected) because I think [the counsellor] seemed to be very helpful, and they were very at ease with me and things*

*like that. So I found it, easier to do than what I anticipated. . . I expected it to be really, really difficult, and really hard. But I, I think because—because of their help, it made it—it did make it easier.*

Participants described how helpful and important it had been to them knowing that their counsellor had experience of losing weight using the TDR programme. Simon compared it with his prior experience of group weight management, saying that the leaders of such behavioural support groups had only ever lost a very small amount of weight, but the TDR counsellor had lost a large amount of weight. He said "I think it helps when you've experienced things rather than just teaching them", and this perception was also emphasised by other participants:

*Jack: I though well. . .it just made me realise that she probably knew what she was talking about, and she's experienced it herself, which made it more sort of genuine really.*

Diane stated that she found out how much weight her counsellor had lost (and maintained) through looking at her social media posts, and felt that the counsellor should have made more of this,

*Diane: -it would have been a bit more encouraging to know how much she had not only lost, but maintained off, you know?*

In addition to losing weight themselves, participants reported that knowing that the counsellor was successfully supporting other people to lose weight was helpful and encouraging.

*Jess: Well they. . .. they talked about—they didn't name people, but they talked about other people's experiences. They'd lost some weight on the same programme, so I felt that they were very knowledgeable about it. And they were able to encourage me quite a bit, actually. And give me sort of success stories, I suppose it was.*

Participants reported how the counsellor's knowledge of dealing with common difficulties helped them to make action plans. Almost all participants were advised in advance to drink a lot of water to avoid constipation. Bernard found the TDR products became bland was encouraged to add flavoured powders to water and herbs to soups. He said that this met his cravings and he found really helpful. Margo was advised by the counsellor to add ice to the shakes so they were more cooling in the summer, and this way they also took a longer time to drink:

*Margo: . . .also they'd say like, like in the summer . . ., make it so that it's a nice long cooling drink. And you know, with plenty of ice, and. Just to make it more satisfying, more nice*

### Time to build a personal relationship

All participants highlighted that their relationship with the counsellor was fundamental to their positive experiences of the TDR programme. Participants' particularly highlighted that counsellor's dedicated time during the sessions helped to build a personal relationship, and this was very important to achieving success.

*Diane: But you know, it's just to get to know 'Susie Smith' as opposed to 'Susie Smith the TDR rep'.*

Many people visited the counsellor at their house, or the counsellor came to their home. They were also in regular sms and telephone communication with the counsellor throughout

the intervention. Participants stated that this was really helpful, and made it easier for them to engage than a group programme at a fixed weekly meeting.

*Jess*: *So, a group session was just not anything that I really could cope with.*

*Margo*: *I think because—with [the counsellor] it's personal. It's one to one. And with like other things, it's big classes. -*

This one-to-one support provided by the counsellors in a personalised support was important to people taking part in the study. Participants described the relationship they established between themselves and their counsellor. They reported that they felt their counsellor really cared about them 'as a person' opposed to just a client, and they really cared their weight loss goals and outcomes.

*Simon: And so they're—they were interested in me. This is, this is the—I found was the difference. . . they appeared genuinely interested in me as a person, and my losing weight. Whereas with [weight loss groups in the community], as I say, I just thought I was there to make the numbers up and give them £5 a week.*

Most people spoke about the relaxed nature and the comfortable and laid-back environment during their sessions with their counsellor. They often said that during the sessions they would talk about things other than weight loss and the TDR programme, such as shared experiences, family, pets, and holidays which helped to build a personal relationship.

*Diane*: *They told me what she was doing; I told her what I was doing. You know, family on holiday, cat's doing this, you know and her—their grandchildren, and yeah, all sorts of things.*

Appointments with counsellors were around half an hour a week but as well as these weekly in-person meetings, participants described that their counsellor called and texted them to send supportive messages, particularly in the earlier stages of the TDR programme. Many patients recognised the amount of time counsellors had available to spend with them contributed to the level of support they experienced.

*Bernard: And they used to text me, and things like that, to sort of encourage me with different things, and things like that. And sending me messages, and things like that, which was—you know—the early stages was really good, really helpful.*

Counsellors also invited patents to call them or text them 'anytime', particularly if they were having a difficult time.

*Margo: Oh, lots of support, yeah. It could be twenty four hours if you wanted. But yeah, they were there. Every Wednesday we'd go—we went to see her. If there was a problem we could just—they'd just say like, "Right, text me if there's a problem, or." Yeah. They were there on hand all the time.*

Diane directly compared the support she received from the counsellor with previous GP or nurse care, and highlighted that the amount of time the counsellor was able to dedicate to her as a key difference between the two.

*Diane: The TDR counsellor has a lot of time for me. And you know when you're going through the practice nurse or a GP in particular, actually, how limited they are in time*

## Adhering to the TDR programme

During the intervention, people were asked to replace all their usual meals with formula food products providing 810kcal a day. Participants reported that initially starting the TDR was difficult because of the sudden transition to a mostly liquid diet. However, they stated that things became easier as they continued the intervention. The lay counsellors facilitated this process by advising participants to focus on the positives and adapt to the negatives, using embracing and avoidance techniques.

*Margo: At the beginning, it was very difficult. At the very beginning. I mean, it was like going through withdrawal symptoms, and stuff like that. And like when you were cooking, I couldn't sort of like taste anything because trying to put it in my mouth to—you know—to test things, just to make sure that the family's food was okay. But then after that, it just came really easy. Very easy.*

Participants acknowledged that some aspects of the TDR programme were hard. This included not being able to eat solid food; difficulties when socialising and cooking for family, being unable to eat what they had cooked; and dealing with holidays. Participants reported either accepting these negatives or adapting their behaviours though use of distraction or avoidance techniques to mitigate the feelings they were experiencing, or expected to experience.

*Simon: There's a lack of social activity, going out with friends and that. Because it—That makes it sound as though I'm out every bloody night wining and dining, I—I can assure you I'm not, but . . . - I personally found it a little bit difficult for a while. But I adapted.*

One participant invited their friends over, instead of going out with them, cooked everyone a TDR product meal. Another took their TDR meals with them on holiday, which they said 'drove' them to stick to the plan. Margo cooked for her whole family every night. Although she could not eat what they were cooking, they said that they adapted to the situation by distracting themselves by eating in a separate room during the early phases of the programme. However, this seemed to be a transient period and, in the latter phases they said were able to sit with her family whilst they ate meals, whilst she stuck to the TDR products. She "didn't really worry" about eating something different:

*Margo:..because what I do is I do their dinner, and I'll go in another room. And then eventually when I'd been on the plan a long time, well for a while, I'd sit with them and my shake and they'd eat, and it didn't really worry me. I was quite content with what I'd got.*

These adaptations were sometimes suggested to participants by the counsellors, and sometimes were novel strategies developed by participants themselves.

One negative experience that almost all participants stated they expected before starting the TDR programme, was to feel hungry. However, when actually taking part in the TDR programme none of the participants reported that they felt hungry.

*Diane: I fully expected to be hungry, to want food. And I just—I didn't even want it. Didn't even want to taste it. It's amazing, really.*

All participants reported wanting drastic weight loss, and presented the positives of the TDR programme as strongly overshadowing any negative they experienced. People seemed to both expect and welcome the more negative aspects of the TDR programme, as they felt the 'severe' nature of the TDR plan would help them to meet their goal of drastic weight loss. Participants frequently spoke positively about the 'strictness', of the programme, which was different to diets they had tried before.

*Bernard: . . . it was quite drastic and severe. So yeah, that's what—I was quite, really happy that I got that*

*Felicity: I thought well, I know what I'm like, it was a case of had to be quite strict diet for me because I can easily slip, so I like the strictness, I liked that you can only have this, this and this, rather than, oh you can have that, and if you wanted a treat you could have this, I actually preferred the strictness of the diet.*

Participants reported that they felt that they "would have lapsed" on other diets, but on TDR they knew what they would be eating from day-to-day and that helped them to stick to the programme. Jasmine reported that the simplicity of knowing what she would be eating day to day helped her fit the TDR into her busy lifestyle. Other participants also said that the TDR programme was easier and 'less of a faff' than previous diets because it was so prescriptive.

*Christine: I had tried various means of losing weight, not very successfully. And I have to say I wasn't all that disciplined. And I got invited to be a part of it, and I thought it was a good opportunity in a supervised environment perhaps to kick start me along, and get me actually on the road to losing the weight I needed to lose."*

## Adverse effects

One of the difficulties highlighted by many participants was constipation during the TDR programme. Many people followed the TDR counsellor's advice to drink more water to manage this problem, and very few presented this as a problem to continuing, or as a particularly negative experience during the TDR programme. For example, despite having to visit the hospital due to constipation, this participant stayed on the TDR programme.

*Simon: But when I became a little bit bunged up and couldn't go—wanted to go to the toilet but couldn't, I—I got a bit worried about that.*

*Interviewer: Okay. And you went to the community hospital, you said?*

*Simon: That's it. I dialled whatever the number is—111 or 101, I can never remember. [laugh] But I- they sent me down the community hospital. And the guy down there was brilliant, he—he gave me—I think he gave me a jab anyway. And from that day on, I never looked back.*

## Outcomes and results

All participants reported intrinsic feelings of wellbeing, which were described as 'feeling good in myself' or 'feeling better in myself' after completing the weight loss phase. The stories they told about these intrinsic positive feelings related to aesthetic changes as well as changes in health, together these contributed to 'feeling good' overall.

Many participants reported that they no longer felt out of breath in their day-to-day activities, and some reported that they had less joint pain. Many people described feeling more

energetic, and, contrary to expectations, they felt more energetic whilst taking part in the TDR programme as well.

*Diane: I had so much energy. Which was amazing.*

Every participant interviewed who had diabetes had their medication stopped or significantly reduced. Diane, who had been taking medications for type two diabetes before taking part, reported that following the trial she no longer needed any medications to manage this condition:

*Interviewer: What are you on now, for example?*

*Diane: For the diabetes?*

*Interviewer: Yes.*

*Diane: Nothing.*

*Interviewer: Oh, right. Okay.*

*Diane: Yes. [laughing] Don't need it.*

Bernard became eligible for a joint operation due to the amount of weight they had lost through taking part in the trial.

*Bernard: [The hospital told me]- ". . . there's no way we can turn you down for an operation now, because you've lost so much weight." So, which was—That was good, as well. So, that contributed to me having been picked for the operation, as well.*

Descriptions of aesthetic changes were commonly related to clothes, and the ability to buy and wear clothes 'off the peg', or to fit into old clothes again. One participant described how this boosted their confidence and contributed to feeling a lot better in themself.

*Bernard: The good thing is, is just going to buy clothes off the peg. I mean, that was—I haven't done that for years. I mean, that was so good. I could just go into a shop, and—and buy a pair of jeans. And like I say, I haven't—I can't remember the last time I done that. And that was such a confidence booster, that sort of thing. And it's like getting shirts that I'd had as birthday presents, and they've been in the wardrobe—real nice shirts, and things like that. And getting them on. I could fit into them. And things like that. Which is, you know—it's all just makes you feel good. In yourself, you feel a lot better.*

## Weight loss maintenance

Participants reported wanting to lose large amounts of weight before starting, and all participants we spoke to did describe significant weight loss success. They also reported that they were keen to keep weight off, and prevent the weight regain that they had experienced after competing other diet programmes.

The most common strategy participants reported for maintaining weight loss, was to continue with the TDR programme. Some people reported paying to stay with their TDR counsellor, whilst others said they would use the products occasionally to maintain weight loss, but not pay for the full programme.

*Margo: I felt um that twelve weeks wasn't long enough. I felt it should have gone on a little bit longer. So we—Because we—I, I didn't actually get to my goal. I had to um say to Carol,*

*"Right, can I carry on with you, and do it like private?" So that I could carry on to get to my goal.*

*Christine: One is I'd probably pull a couple of the shakes out of the box. . ..They're somewhere in the cupboard. And so I'll have that instead of lunch. The other one is just the—the approach. Some of the, some of the recipes, some of the, the things that you make. Because the recipes give you ways of making things without certain ingredients, so that they're less fattening, they're less with, you know, with fat.*

However, the TDR had been provided free of charge during the DROPLET trial, and some participants reported that even though they wanted to lose more weight, and wished to continue with the TDR, cost was a barrier. One participant found a TDR product for sale on the internet that cost less than the TDR programme offered in the trial, but did not include the behavioural support and contact with a counsellor.:

*Bernard: I've found another product. On the, on the internet. It's called [discount brand TDR]. And it's basically very similar to the [named brand TDR]. It's the shakes, the bars, the meals. But it's—it's a lot cheaper. Because the [named brand TDR] is—Don't get me wrong, it's served its purpose and it was brilliant, but it can be quite expensive. When you're paying it out, you know, sort of sixty odd pound a week, or more, it can work out quite expensive.*

Of those that did pay to continue, one participant reported that she was more likely to cheat, because she was now paying for it herself, rather than receiving free products as part of the study, and another that she was less motivated to see her counsellor.

*Jess: Sometimes I do cheat a bit, I must admit. . . But I figure that—I suppose when I was on the, on the study, I felt duty bound to keep to it. Now I'm actually paying for it myself [laughing], I feel that sometimes I can . . . give myself some slack sometimes, you know.*

*Jasmine: Knowing I've got to pay for it and organise it myself is a bit of a demotivator as well.*

There was an appreciation for the education about weight control that was provided on the programme, as well as the formula food products themselves, which were helpful when maintaining weight loss. Participants described how support received from their counsellor during transition back to 'real food' encouraged them to think differently about portion size, activity levels, and their overall approaches to food and eating. They reported that they would continue to use the skills from the lessons now that they mostly eat normal food.

*Christine: Because the lessons learned, the change in the way I approached my eating, the way in which I thought about it. . . Whereas in the past you might have said, 'oh well, you know, I've worked in the garden today—that's great, I'll have an extra potato'. You know, you think about it slightly differently.*

The strategies used to maintain weight loss including continuing to follow the intervention, or use aspects of the diet programme, and implement learning about food and eating, contributed to people reporting confidence that they would not regain weight, and would know what to do if they did notice some weight gain.

*Christine: I now know that I will never—I will never put that weight back on again.*

*Bernard: Changed me, changed my life.*

### Recommending TDR to others

Every participant interviewed was keen to recommend the TDR to other people. They emphasised the benefits that they received from the TDR programme and stated that other people could experience these too. Participants associated their reasons behind wanting to recommend the TDR programme to others with the benefits they had experienced.

> *Felicity: Just, if anybody's ever thinking, "should I do it", don't, don't question it, do it, it's well worth it. The health benefits, your, your whole personality, your self-confidence, it just builds everything, because your, you see, what it is you wanted to be.*

## Discussion

There is now good evidence from several recent clinical trials that low-energy total diet replacement weight loss programmes are one of the most clinically effective non-surgical weight loss treatments in the short to medium term [1–3]. In contrast to the clinical trials, which provide information on the clinical outcomes, qualitative studies, typically embedded within trials provide a valuable opportunity to explore the perceptions and experiences of the trial participants of the interventions being tested. In this case, the verbalised experiences of the DROPLET trial participants will go some way to determining the suitability of TDR weight loss programmes as an option for the routine treatment of obesity.

Generally, the results of this study suggested that the TDR weight loss programme was acceptable, relatively easy to follow, and the participants interviewed were overwhelmingly pleased with the results and all participants were keen to recommend the programme to others.

One distinction between this study and other studies of TDR programmes is the method of delivery of the TDR intervention. Other studies have utilised physicians or healthcare professionals to deliver the TDR intervention in combination with medical monitoring [1, 3]. However, the DROPLET study tested an alternative delivery model whereby the intervention was delivered by a local lay counsellor in the community. These counsellors are managed by an established TDR provider (Cambridge Weight Plan UK Ltd) who offer the programme direct to the public. Some participants had heard of the TDR approach and brand name before, some had endorsements from family or friends which acted as a stimulus to engagement for some participants. Public awareness of TDR programmes has increased, particularly since the results of the DROPLET and DiRECT trials have been published [2, 3]. It is plausible that as this approach becomes more widely known acceptability and uptake may increase.

The support and guidance, and the relationship that participants built with the counsellor who delivered the TDR programme was central to many of the positive themes emerging from the interviews. It was clear that the flexibility of contact was valued, and the informality of the interactions allowed participants and counsellors to build a relationship beyond that of counsellor and client, which is distinctive to the formal relationship between healthcare professional and patient [17].

Participants reported several different reasons as their motivation for taking part in this study. They were aware that they were overweight, but also perceived that through losing weight they would likely see improvements in their health, appearance, self-esteem, mental well-being. This is consistent with previous studies that have reported that motivation to embark on a weight loss attempt was facilitated by medical concern, clinician endorsement, and dissatisfaction with current appearance [18–24].

As in previous studies, the majority of those interviewed reported undertaking (several) previous weight loss attempts [15, 16, 25]. These previous weight loss attempts had been

unsuccessful, or if they had managed to lose some weight in the past, they had reported that over time they had gradually regained any weight previously lost. Most often, previous weight loss attempts included attending community weight-loss groups. A 12 week referral to such programmes have been shown in clinical trials to achieve approximately 3kg weight loss over a 1 year period [26, 27] and its' possible that slow weight losses were one of the reasons that participants believed their past weight loss attempts had been unsuccessful [16, 25].

The outcomes that participants reported included increased physical and psychological wellbeing, and these findings align with the quantitative findings reported in the main outcome paper [2]. Participants themselves did not often distinguish between physical and psychological health and described intrinsic feelings of wellbeing, which encompassed both of these categories, showing that physical health and psychological wellbeing are deeply intertwined. Future studies should bear in mind that 'wellness' for patients is a holistic rather than dichotomised concept.

The support provided by one-to-one counsellors and the fast results were key in motivating adherence and success, and differentiated this programme from previous weight loss attempts. Being part of a clinical trial offered some participants added accountability, which is consistent with some previous qualitative research on low energy diets [15, 16].

A notable feature in the present study was that participants repeatedly emphasised the role of and relationship with the counsellor in contributing to their continued adherence and success. Knowing the counsellor had themselves lost weight using the TDR programme, and had experience of working with other individuals who had lost weight was extremely important to the participants. The participants felt the counsellors had knowledge and deep understanding of their experience which contributed to the empathy they displayed. A meta-analysis of studies exploring the effect of psychotherapy therapist empathy on client outcomes, reported that the level of empathy displayed by a therapist is a moderately strong predictor of therapy outcome [28], and future studies could explore in more depth the importance of empathy and embodiment in participant/counsellor relationships.

Previous studies report that individuals feel that there are many personal, social and environmental barriers associated with their ability to, engage in weight loss programmes, these include a lack of time, lack of motivation, partner support and/or knowledge on how to enact behaviour change [29–31]. Here the participants described the time the counsellors spent in face-to-face and remote counselling sessions as a key factor in supporting them to both stick to the diet, and to cope with difficult moments. Furthermore, although participants may perceive that the amount of time they spent with the counsellor helped them lose weight, there is evidence to suggest that the actual time spent is not as important as what happens during a consultation when predicting outcomes [32]. Relevant to this, the participants highlighted the counsellors knowledge and enthusiasm as key factors that aided their motivation.

A previous study that aimed to understand the patient experience of using TDR treatments, where the support was provided by healthcare professionals, reported that partner support was perceived as key in motivating diet adherence [33] but this was not evident from our interviews. It is possible that because the level of one-to-one support provided by the TDR counsellor was frequent, including remote interactions and contact and/or appointments scheduled at evenings, weekends, and times outside traditional working hours, the relationship with and support provided by the counsellor replaced the need for support from a partner or family member. Whist other studies have explored clinician-mediated support this study shows that success on a TDR can be effectively facilitated by a lay-counsellor and this is both effective and well received by patients.

The TDR weight loss programme was delivered free of charge as part of this study. Interviews highlighted that this was important to participants, since cost was considered a barrier.

However, perhaps counterintuitively, they speculated that they might be less likely to adhere to the diet if they had paid for it themselves. A recent trial in which doctors encouraged patients to pay to attend a weight loss programme which they needed to self-fund showed minimal uptake, whereas the same programme was attended by 40% of participants were treatment was funded through the healthcare system [34]. This aligns with existing work showing that although providing free of charge statin prescriptions to patients following myocardial infarction does not affect uptake of the treatment, providing the treatment free of charge does has a beneficial effect on treatment adherence compared with providing the same treatment through co-payment or cost-sharing schemes [35]. This an important aspect when considering wider implementation of this programme.

Strengths of this study include the use of semi-structured interviews allowing flexibility for participants to describe the aspects of the experience that were important for them. A further strength was that interviews were conducted by people with detailed knowledge of the components of the trial, including the timelines participants were following as they moved from TDR into normal food, and could understand the terminology used by participants, although participants did not know the interviewers.

The greatest limitation to this study is that the people we interviewed lost more weight on average than people randomised to the TDR programme in the trial. Although weight loss at 1 year in the sample ranged from an overall loss of 32kg to a gain of 2kg, these people may have had more positive experiences of the TDR programme. Recruitment bias is common in this and other qualitative studies related to weight management presumably because people who were not as successful, or those who drop-out may be reluctant to take part in interviews. Participants in this study were recruited on the basis of their baseline demographics, therefore we were unable to select based on weight at 1 year, as that outcome was not known at the time of recruitment. However, although the sample was not representative of the broader population, it did reflect the demographic profile of the population taking part in the trial. To support wider implementation of TDR programmes, future studies should take a maximum variation sampling approach across the wider population, and could explore if and how aspects of participants lifeworld influenced their ability to engage positively with the TDR (including the role of food, relationships, family structure, and cultural factors). It is possible that participants' responses are systematically biased toward respondents' perceptions of what is "correct" or socially acceptable, otherwise known as social desirability bias [36]. Since we conducted the interviews as part of a study that involved many parties, the participants may have been conscious not to say anything that might offend or upset any of these related parties (e.g. GP's counsellors, researchers), but we aimed to mitigate this by clearly stating all comments would be confidential and not shared with counsellors or GPs. All interviews took place after the participants had completed the weight loss phase, when most participants were no longer following the intervention. The limited reporting of negative experiences may be due to recall bias, focussing on positives they are experiencing currently, rather than negatives that they had experienced in the past [37]. Future studies could conduct a series of interviews with each participant over the course of the intervention to capture how participant experiences might change during different phases.

In summary, this qualitative study of the participants taking part in the DROPLET randomised controlled trial suggests that the delivery of a low-energy TDR behavioural intervention through one-to-one support provided by lay counsellors in the community is highly acceptable to participants. Furthermore, this method of delivery may offer some benefits to participants over more traditional health professional-led programmes.

## Conclusion

The referral of patients by their healthcare professional to a TDR weight loss programme delivered by lay-counsellors in the community could be considered a suitable alternative to the delivery of TDR by healthcare professional in a medical setting. This TDR programme was well received by patients, who welcomed the programme, in spite of its restrictions. They felt that it differed from previous weight loss programmes they had tried, and wanted to engage with this programme because of the potential to lose a substantial amount of weight. Participants emphasised the key role of the lay-counsellor in supporting successful weight loss, and their overall positive feelings during, and after completing the programme.

## Supporting information

**S1 Checklist.**
(DOCX)

**S1 Material.**
(PDF)

## Author Contributions

**Conceptualization:** Nerys M. Astbury, Susan A. Jebb.

**Data curation:** Nerys M. Astbury, Charlotte Albury, Rebecca Nourse.

**Formal analysis:** Nerys M. Astbury, Charlotte Albury.

**Funding acquisition:** Nerys M. Astbury.

**Project administration:** Nerys M. Astbury.

**Supervision:** Susan A. Jebb.

**Writing – original draft:** Nerys M. Astbury, Charlotte Albury.

**Writing – review & editing:** Nerys M. Astbury, Charlotte Albury, Rebecca Nourse, Susan A. Jebb.

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
