## [Decision Letter · Decision Letter 0]

17 Jun 2020

PONE-D-20-05095

Participant experiences of a low-energy total diet replacement programme: a descriptive qualitative sub-study of participant in the Doctor Referral of Overweight People to Low-Energy total diet replacement Treatment (DROPLET) trial

PLOS ONE

Dear Dr. Astbury,

Thank you for submitting your manuscript to PLOS ONE. After careful consideration, we feel that it has merit but does not fully meet PLOS ONE’s publication criteria as it currently stands. Therefore, we invite you to submit a revised version of the manuscript that addresses the points raised during the review process.

We look forward to receiving your revised manuscript.

Kind regards,

Louisa Ells, Ph.D.

Academic Editor

PLOS ONE

Additional Editor Comments:

This is an important piece of research that is worthy of publication but requires some substantial revisions to address some key methodological, interpretation and quality issues as identified by the reviewers. We recommend that authors use the COREQ checklist or other relevant checklists listed by the Equator Network such as the SRQR to ensure complete reporting (http://journals.plos.org/plosone/s/submission-guidelines#loc-qualitative-research), and would expect qualitative studies to include the following: 1) defined objectives or research questions; 2) description of the sampling strategy including rationale for the recruitment method participant inclusion/exclusion criteria and the number of participants recruited; 3) detailed reporting of the data collection procedures; 4) data analysis procedures described in sufficient detail to enable replication; 5) a discussion of potential sources of bias; and 6) a discussion of limitations. Addressing the reviewers comments will ensure you meet these requirements.

2. When reporting the results of qualitative research, we suggest consulting the COREQ guidelines: http://intqhc.oxfordjournals.org/content/19/6/349.

In this case, please consider providing the interview script used; and please discuss whether bias issues were considered.

4. Thank you for stating the following in the Financial Disclosure section:

'NMA and SAJ are supported by funding from the NIHR Oxford Biomedical Research Centre (BRC). RN and SAJ were supported by funding from NIHR Applied Research Care (ARC) Oxford. SAJ is a NIHR) senior investigator. The main DROPLET trial was funded by a research grant from Cambridge Weight Plan Ltd UK to University of Oxford. We conducted this sub-study independently from the main trial, with separate funding from National Institute for Health Research (NIHR) Applied Research Care (ARC) Oxford. The funders had no input into the design, data collection, analysis decision to publish or preparation of the manuscripts. The authors had a right to publish regardless of the results. The views expressed are those of the author(s) and not necessarily those of the NHS, the NIHR or the Department of Health and Social Care.'

We note that you received funding from a commercial source: Cambridge Weight Plan Ltd UK

Reviewers' comments:

Reviewer's Responses to Questions

**Comments to the Author**

1. Is the manuscript technically sound, and do the data support the conclusions?

Reviewer #1: Yes

Reviewer #2: No

2. Has the statistical analysis been performed appropriately and rigorously? 

Reviewer #1: N/A

Reviewer #2: N/A

3. Have the authors made all data underlying the findings in their manuscript fully available?

Reviewer #1: Yes

Reviewer #2: No

4. Is the manuscript presented in an intelligible fashion and written in standard English?

Reviewer #1: No

Reviewer #2: Yes

5. Review Comments to the Author

Reviewer #1: Dear Authors, thank you for submitting this interesting paper. Overall, I think some clarifications and amendments are required to strengthen your manuscript. Please find these listed below:

* Please proof-read the manuscript; there are various typos across the document.

* Please check the Consolidated Criteria for reporting Qualitative Research (COREQ) - this aims to promote complete and transparent reporting to improve rigour and credibility of findings. Based on the review of your manuscript and the checklist, the following amendments are needed:

* The credentials and the gender of both interviewers were not reported.

* The authors did not mention what the participants were told about the interviewers. Were they told about their personal goals?

* What were the interviewer characteristics? This relates to reflexivity

* There needs to be further explanation regarding the sampling, as there was no mention of how many people were sent the letters and how many refused.

* Although the demographics were reported, there were some characteristics that were missing, such as co-morbidities, medications taken and stopped. it would also be useful to report on the counsellors, as it was not clear how many different counsellors dealt with the 12 participants and if this had an effect on the findings.

* More information is required regarding the topic guide, such as how it was developed, was it pilot-tested, were prompts and probes used, if the guide was amended throughout the interviews, were there discrepancies between interviewers as 2 people conducted interviews and how was this dealt with, etc.

* The topic of data saturation was not mentioned at all. Were the 12 just chosen based on those who responded and no further participants were needed? At what point was data saturation reached?

* Were transcripts shared with participants and checked by them?

* The data analysis section needs to be clarified; who exactly coded the data and how, describe the coding process, how were themes derived, techniques used to enhance trustworthiness and credibility of data analysis

* Quotes need to be identified as at the moment we are unaware of who said what.

* The explanation of the lay counsellors background needs to be in the introduction rather than the discussion

* Please explain why participants were not interviewed face-to-face

* Limitations should also include the fact that the sample was not diverse in terms of ethnicity and gender

* It would have been interesting to include those that dropped out to identify the barriers towards this intervention. As you mentioned in the limitations, the participants interviewed only had positive feedback, which does not only relate to recall bias as you said.

* You mentioned in the discussion that a formal relationship (such as with a HCP) may not have provided similar acceptability for the intervention. The relationship with HCPs may play a different role and should not be disregarded. Some people may actually adhere to such an intervention if prescribed by a HCP due to trust. I think the main factor here was the availability of support and the fact that the counsellors had undergone a similar experience, thereby participants were seeing a live example and felt reassured.

* It would be worth mentioning the level of support provided by each counsellor to each participant. This may have had an impact on findings and should have been reported and explored.

Reviewer #2: This research attempts to address an important need within the area, and qualitative approaches need to be utilized to answer the questions unanswered in this area. The premise for this study is therefore strong. The writing is also of a good quality, with very few obvious grammatical mistakes. However, there are a number of problems with the research design and subsequent claims that are being made.

Line Specific Comments:

Line 57; was not as

Line 58; Total diet replacement abbreviated to TDR in Methods but not in intro. Abbreviated in abstract. Looking back on intro it is written both ways. Consistency needed.

Line 60; Does HRA need writing out before using an abbreviation?

Line 100; sentence needs reviewing; programme added one to many times

Line 117; ‘they’ is missing

Line 147; what other studies?

General Comments:

It would be useful, and necessary to say something about the assumptions this research is rooted in. Not stating these hinders the readers ability to fully understand how the authors have arrived at the knowledge in this paper.

The use of the term ‘sub-study’ seems misguided; it does a disservice to what is ultimately important research and perhaps only tells the reader something about the authors. ‘Sub-study’ suggests that this study is, in the authors view, of ‘lesser’ importance. I appreciate that the study population has been sampled from a larger cohort, but that does not condemn this research to ‘sub-importance’. This study should set out with different assumptions to answer different questions, and with regard to a contribution to knowledge this study should be of equal importance. I therefore find the use of the term ‘sub-study’ as an act of self-depreciation. Importantly, it seems entirely unnecessary. However, the use of ‘sub-study’ does suggests something about how the authors have approached this work and the paradigm they are orientated in. Given that the authors do not outline the assumptions this research is based on (see above comment), the reader can only make these connections. I suggest the use of this term should be re-considered, or the authors should say something about their assumptions that render the use of this term as being congruent.

Sampling; this section is titled sampling strategy but lacks any details of how a strategy was deployed. The section heading is therefore misleading. Purposive sampling is a broad umbrella term that captures different types of purposeful sampling, such as a total population sampling or a maximum variation sample. These ‘types’ might give an indication of a strategy. The authors go on to state that they aimed to obtain a ‘similar balance’ as the main trial but provide no detail of what this was or if this was achieved. Sampling this ‘similar balance’ is problematic for a number of reasons. First, it perpetuates the potentially already distorted sample from the DROPLET trial. Do socio-economic and ethnic groups respond in equal measure to a letter from their GP? An effective purposeful sampling strategy would have considered who was part of the DROPLET trial and applied purposeful sampling that was mindful of the sample rather than further distorting it. Second, aiming to achieve a ‘similar balance’ in this ‘sub-study’ again tells the reader more about the authors assumptions than sampling itself. It suggests the authors assume that a random sample is appropriate, and that there are realist assumptions at work. This therefore does not speak up for the inequality in health, or the power that researchers hold in knowledge production. This section, at the very least, needs to include detail of who was in the DROPLET trial – and therefore the constraints on sampling in this study – and what strategy was used in the current study and why. This is necessary as this underpins the claims that are subsequently made.

Sampling two - In the Analytic Approach section; the claim that “the end goal was to understand how the acceptability of the programme and the potential for it to be offered at scale” is troubling. For this to be achieved, a sampling strategy that ensured this research represented a broader population, or the population that TDR is aimed at would have to be deployed. As things stand, the knowledge produced in this paper only speaks to the experience of middle-aged white people (we can also make reasonable assumptions about the socio-economic groups that are likely to be represented in the DROPLET trial). This is not to say that these views are not important, but this research is limited in how it can inform any ‘scaling up’.

The use of the term ‘recruitment bias’ on line 596 further suggests realist assumptions are at the base of this work and raises further questions over the suitability of sampling.

Interviews; if a semi-structured interview guide has been based on the experience of the team it would be useful to understand what this is. The knowledge produced in this research is dependent on the researchers interpretative practices. The researchers experience cannot be separated from these practices and good qualitative research would say something about the researcher as much as interviewees. Hiding the researcher, and/or the researchers experience away is cognizant with the objectivity strived for in research that has realist assumptions.

Quote; I is not necessary to write ‘um’ in quotations. It means quotes do not read well and does not necessarily change the meaning when removed. As long as the meaning is maintained, I would recommend deleting the ums.

Reasons for taking part; there is no references in this section, which seem odd. There is a lot of literature that would support these findings; 1). Weight-loss being prompted by wanting to look a different way, 2). Weight-loss being a lifelong thing, 3). The importance of the doctor in getting people to a programme. References to support these findings, and to demonstrate that these findings are not new or unique should be considered.

The quotes in this section are long considering the points they are supporting are descriptive in nature. Given the suggestion to add text elsewhere, text can be saved here.

Support and guidance from the counsellor and time to build relationships; again, these findings are interesting but not uncommon. Reference to literature that talks of the importance of the counsellor should be added. These findings are largely descriptive, and while interesting, offer little insight into the importance of the counsellor. An analysis of the importance of empathy and embodiment would provide greater insight into why this programme worked for these 12 interviewees (and why it might not for others). It would be useful to clarify if the counsellor is a paid member of staff or a volunteer, as well as the lay counsellor recruitment (was it necessary they had experience of losing weight?) and their training (were they trained on being empathetic, dealing with a variety of groups etc). The counsellor is obviously an important part of this process, as the literature would suggest, so understanding a bit more about them would strengthen an understanding of why DROPLET worked for a group of white middle-aged adults. Some of this comes in the discussion but would be useful prior to presenting the findings.

These sections effectively highlight the importance of the counsellor but do nothing to critique the bases for this relationship in the current context. Given that there are aspirations to scale up the DROPLET programme, reporting these findings uncritically has the potential to do more harm than good. The questions that remain unanswered here are; what is the context of these individuals that enable them to receive these relationships positively, how the sampling of this study precede these findings, what are the cultural factors at play, how are the lives of these interviewees structured, what place does food play, how does ideology and discourse feature. Downstream programmes that draw on individual responsibility have been shown to widen inequalities. If research like this remains uncritical it only exacerbates and perpetuates this widening and implicates researchers as part of the problem rather than the solution.

Following the TDR programme; some of the above point are briefly explored here. On first reading I assumed there was a temporal reference in this theme, but on reflection understood it as adhering to the programme. Maybe consider changing this to avoid others making the same mistake.

Themes adverse effects and recommending TDR to others; it’s hard to see what these themes add. They are descriptive and overly simplistic.

Acceptability; I have issues with the use of this term. This paper effectively shows that 12 white middle-aged interviewees had a positive experience and outcome on a TDR programme. It does not say much, if anything about how and why it is acceptable to this group, or indeed broader social groups (which is necessary for scaling up). An understanding of acceptably can only come by considering the broader social and cultural context. Whether you draw on practice approaches in sociology, or dual-processing theories from psychology, we know that a large part of human action is automatic, unconscious, and beyond rational and cognitive reasoning. I fail to therefore see how the authors can claim that acceptability can be determined from this research. The analysis in this paper is too simplistic to and descriptive to make these claims. The fact the authors have used ‘will go some way’ in line 506 suggests they know or feel that they have made a leap from talking about the descriptive experience of a narrow group of interviewees to talking about acceptability. However, at it’s core I agree with the authors intentions, and qualitative research has an important role to play in the evaluation of these programmes.

Discussion; large parts of the discussion provide a similar level of description to the findings, and I therefore feel as though I am reading the same thing again. However, this time there is literature that was missing in the findings.

One of the stronger parts of this paper is the section on addressing the limitations. However, it reads as if it was written by somebody else as an afterthought. Some of the points here are critical

Overall comments; the most compelling part of this paper is the content about lay-counsellors. I would recommend focusing more on this topic by moving beyond description and providing some analysis of these relationship that critique the context they are embedded within. It is also necessary that the authors provide a more honest account of their sampling methods and the claims that are made about what this knowledge can do, based on the obvious limitations in sampling. It would then be appropriate to make claims about acceptability (for a very specific group). It is also necessary for this research to set out the assumptions it is rooted in and consider how there is a level of internal consistency throughout.

6. PLOS authors have the option to publish the peer review history of their article (what does this mean?). If published, this will include your full peer review and any attached files.

Reviewer #1: No

Reviewer #2: No

---

## [Author Response · Author response to Decision Letter 0]

18 Aug 2020

Reviewer 1

1. Please proof-read the manuscript; there are various typos across the document.

We have thoroughly proof read the submission, and corrected any errors or typos.

2. Please check the Consolidated Criteria for reporting Qualitative Research (COREQ) - this aims to promote complete and transparent reporting to improve rigour and credibility of findings. 

Each of the points: 3-16 raised by Reviewer 1 relates to items on the COREQ checklist. In order to make it easier for readers to determine where each item of the COREQ checklist are located, we have attached a COREQ checklist to our submission, which details the location in the manuscript (page and line number) where each checklist item is located.

3. The credentials and the gender of both interviewers were not reported.

We have now included details of the credentials and gender of the interviewers.

4. The authors did not mention what the participants were told about the interviewers. Were they told about their personal goals?

Participants were told that the interviewers were aiming to understand the thoughts feelings and opinions of the participants, regardless of whether these were good or bad, and we have no added a statement to the sampling strategy section of the methods to clarify this:

“Participants were told that the purpose of the study was to explore their thoughts and feelings of the experience of taking part in the DROPLET study. regardless of whether these were good or bad.”

5. What were the interviewer characteristics?

We have now included further detail on the interviewer characteristics, and referenced the location of this information in the enclosed COREQ checklist

6. There needs to be further explanation regarding the sampling, as there was no mention of how many people were sent the letters and how many refused.

We have now added additional details in respect to the sampling method in the sampling strategy section of the methods:

“We invited a purposive sample of participants via letter to take part in this qualitative study. We aimed to obtain a sample with similar baseline demographics to the participants in the main DROPLET trial. A total of twelve participants were approached to take part in this qualitative study. None of the people approached to take part in the interviews refused, and none of the participants who agreed to take part dropped out.”

7. Although the demographics were reported, there were some characteristics that were missing, such as co-morbidities, medications taken and stopped. it would also be useful to report on the counsellors, as it was not clear how many different counsellors dealt with the 12 participants and if this had an effect on the findings

We have sought guidance from an information security expert, who has advised us not to include any additional individual-level participant characteristics which, together with the information already provided might enable identification of the participants in this study.

However, we agree with the reviewer that it would be useful to know some additional details of the group interviewed. Therefore we have included additional details on the group level characteristics including the number with co-morbidities, the number of counsellors participants were assigned to and the mean number of sessions attended of the participants who took part in this study in the results section:

“Twelve participants were interviewed (3 Male, 9 Female), all were White British and middle aged (range 40-75 years). To maintain the anonymity of the participants we have used pseudonyms to identify quotes.

All eight of the counsellors who were involved in delivering the TDR programme are represented in the sample. Three participants had a pre-existing diagnosis of type 2 diabetes and two had hypertension at the time they enrolled on the trial. At 1 year the average weight loss in these participants was -15.4 kg (10.3) which is somewhat greater than the average for the whole sample, and ranged from weight loss of 32.3kg to a weight gain of 1.9kg”

8. More information is required regarding the topic guide, such as how it was developed, was it pilot-tested, were prompts and probes used, if the guide was amended throughout the interviews, were there discrepancies between 

interviewers as 2 people conducted interviews and how was this dealt with, etc.

We have now included additional information on the interview guide:

“We developed a semi-structured interview topic guide based on the experience of the team, and the wider qualitative literature on weight management (Supplementary Material). The interview guide was piloted with colleagues prior to its use with participants. The schedule consisted of twelve open-ended questions with prompts to explore participants’ experience of participating in the DROPLET trial and in particular, views on the TDR programme and type of weight-loss support.”

In addition to this additional detail, we’ve also included a copy of the interview guide as supplementary material to the submission.

See also response to Reviewer 2 (Comment 6)

9. The topic of data saturation was not mentioned at all. Were the 12 just chosen based on those who responded and no further participants were needed? At what point was data saturation reached?

We have now added additional information with regard data saturation:

“Regulatory approvals were in place to interview up to 30 participants, but data saturation was reached after 10 participants, and we interviewed an additional 2 participants to confirm this.”

10. Were transcripts shared with participants and checked by them?

We did not share the transcripts with the participants for checking, and have stated this in our methods section:

“Interviews were transcribed verbatim, transcripts were not shared with participants. This was to reduce participant burden as they were taking part in a large trial.”

11. Quotes need to be identified as at the moment we are unaware of who said what.

To protect the anonymity of the participants, but addressing the issue of identifying the quotations, we have created pseudonyms for the participants and used these to identify all quotes in the manuscript. We have now clarified this in the methods, and used the pseudonyms to identify all quotes:

“To maintain the anonymity of the participants we have used pseudonyms to identify quotes“

12. The explanation of the lay counsellors background needs to be in the introduction rather than the discussion

We’ve now moved the description of the background of the counsellors to the introduction.

13. Please explain why participants were not interviewed face-to-face

All the participants in this study were recruited from the people who took part in the main DROPLET trial. This study in itself involved attending a 24 week weight loss programme as well as up to six additional research visits at GP practice, therefore in order to minimise participant burden we took the decision to conduct the interviews over the phone. We have included this justification in the methods section:

“To minimise participant burden all interviews took place over the telephone …”

14. Limitations should also include the fact that the sample was not diverse in terms of ethnicity and gender

We understand that the sample is not representative of the UK population in terms of gender and ethnicity. However, the purpose of the study was to explore the thoughts and feelings of the participants of the DROPLET study. The sample chosen was representative of this population. We have taken care not to generalise the findings of this study to the entire population, and have added clarification throughout the paper that learning from this study applies to the specific population which we sampled.

15. It would have been interesting to include those that dropped out to identify the barriers towards this intervention. As you mentioned in the limitations, the participants interviewed only had positive feedback, which does not only relate to recall bias as you said.

We agree, and we have discussed this issue in the discussion, in relation to the difficulty recruiting people who refuse to take part or drop out of clinical trials. 

In reality the number of drop-outs was small, but we agree with the reviewer a better understanding of the thoughts of these individuals would be useful and interesting. To address this we have now included a statement in the discussion that any future evaluations of TDR programmes delivered in routine care should aim to explore the thoughts experiences and feelings of those who refuse to participate and those who drop out, as well as those who partake in the treatment.

16. You mentioned in the discussion that a formal relationship (such as with a HCP) may not have provided similar acceptability for the intervention. The relationship with HCPs may play a different role and should not be disregarded. Some people may actually adhere to such an intervention if prescribed by a HCP due to trust. I think the main factor here was the availability of support and the fact that the counsellors had undergone a similar experience, thereby participants were seeing a live example and felt reassured

We think the reviewer has mis-understood. Since we did not compare similar interventions delivered by HCP with those delivered by lay counsellors, we cannot suggest that interventions prescribed by HCP are any less acceptable. 

The intervention delivery model used in the DROPLET study used initial prescription/endorsement by a HCP (practice nurse and GP), but delivery of the programme was the responsibility of the lay counsellors in the community. The findings of this study suggest that this model is acceptable to participants, because of a combination of the endorsement of the HCP and the flexible support and less formal relationship with the counsellors (these are components that are distinct from a patient HCP relationship)

We have amended the wording in the discussion to clarify that there is no comparison with HCP delivered methods

“….low-energy TDR behavioural intervention through one-to-one support provided by lay counsellors in the community is highly acceptable to participants. Furthermore, this method of delivery may offer some benefits to participants over more traditional health professional-led programmes”

17. It would be worth mentioning the level of support provided by each counsellor to each participant. This may have had an impact on findings and should have been reported and explored.

We have now included a brief summary of the level of support provided by each counsellor as per protocol, and detail of the mean number of sessions attended by the participants interviewed in this study has also been included in response to commend 7 by Reviewer 1.

Reviewer 2

1. Line Specific Comments:

• Line 57; was not as

Thank you for drawing our attention- we’ve corrected this typo

• Line 58; Total diet replacement abbreviated to TDR in Methods but not in intro. Abbreviated in abstract. Looking back on intro it is written both ways. Consistency needed.

We have now defined TDR at first use in the Introduction and thereafter used the abbreviation. Because the abstract is a standalone section we have defined TDR at first use.

• Line 60; Does HRA need writing out before using an abbreviation?

We have now defined this abbreviation at first use

• Line 100; sentence needs reviewing; programme added one to many times

We have now rephrased this sentence:

“The end goal was to understand the acceptability of the programme for participants in the DROPLET trial”

• Line 117; ‘they’ is missing

We have now included this omission

• Line 147; what other studies?

We have now added references to the studies that have reported that participants feel added accountability when taking part in a clinical trial.

2. It would be useful, and necessary to say something about the assumptions this research is rooted in. Not stating these hinders the readers ability to fully understand how the authors have arrived at the knowledge in this paper.

The use of the term ‘sub-study’ seems misguided; it does a disservice to what is ultimately important research and perhaps only tells the reader something about the authors. ‘Sub-study’ suggests that this study is, in the authors view, of ‘lesser’ importance. I appreciate that the study population has been sampled from a larger cohort, but that does not condemn this research to ‘sub-importance’. This study should set out with different assumptions to answer different questions, and with regard to a contribution to knowledge this study should be of equal importance. I therefore find the use of the term ‘sub-study’ as an act of self-depreciation. Importantly, it seems entirely unnecessary. However, the use of ‘sub-study’ does suggests something about how the authors have approached this work and the paradigm they are orientated in. Given that the authors do not outline the assumptions this research is based on (see above comment), the reader can only make these connections. I suggest the use of this term should be re-considered, or the authors should say something about their assumptions that render the use of this term as being congruent.

We agree, and have reconsidered our use of sub-study, referring to this instead as a ‘qualitative’ study in its own right. We have removed all references to a sub-study, and have rephrased the title removing reference to the main DROPLET trial, to make it clear this is a study in its own right.

We have also explicated our assumptions and paradigm more clearly in the methods section, stating both our ontological and epistemological positioning:

“We followed a qualitative descriptive approach to analysis. Our ontological position was relativism, and our epistemological assumptions were grounded in subjectivism. In following this positioning, we viewed this research as both inductive and subjective. We remained aware of the active role of the researcher in co-creating data as it was generated, and also in influencing interpretation through the process of analysis.”

3. Sampling; this section is titled sampling strategy but lacks any details of how a strategy was deployed. The section heading is therefore misleading. Purposive sampling is a broad umbrella term that captures different types of purposeful sampling, such as a total population sampling or a maximum variation sample. These ‘types’ might give an indication of a strategy. The authors go on to state that they aimed to obtain a ‘similar balance’ as the main trial but provide no detail of what this was or if this was achieved. Sampling this ‘similar balance’ is problematic for a number of reasons. First, it perpetuates the potentially already distorted sample from the DROPLET trial. Do socio-economic and ethnic groups respond in equal measure to a letter from their GP? An effective purposeful sampling strategy would have considered who was part of the DROPLET trial and applied purposeful sampling that was mindful of the sample rather than further distorting it. Second, aiming to achieve a ‘similar balance’ in this ‘sub-study’ again tells the reader more about the authors assumptions than sampling itself. It suggests the authors assume that a random sample is appropriate, and that there are realist assumptions at work. This therefore does not speak up for the inequality in health, or the power that researchers hold in knowledge production. This section, at the very least, needs to include detail of who was in the DROPLET trial – and therefore the constraints on sampling in this study – and what strategy was used in the current study and why. This is necessary as this underpins the claims that are subsequently made.

We are happy to clarify that we did not use a random approach to sampling. Rather ‘similar balance’ refers to our aim to sample a population that represented those taking part in the main DROPLET study, rather than the wider population. We have included more detail of the total population taking part in the DROPLET trial in this section. We agree limitations on sampling are important, and these are presented in detail the ‘strengths and limitations’ section of this paper.

Our aim with this paper is to explore people’s experiences of taking part in the DROPLET trial. Therefore the reviewer’s question do ‘socio-economic and ethnic groups respond in equal measure to a letter from their GP?’ is beyond the scope of what we aim to achieve here and would require a much larger group with more specific questioning about areas . Instead our paper assesses particular participants experience of this particular trial and we cannot comment on the wider inequality in health. 

However we have added clearer acknowledgement of the power that researchers hold in knowledge production. As with all qualitative research we shaped the data by the questions we asked (and did not ask), and in our analysis and interpretation of the data. This was something of which we remained aware throughout the process, and now explicitly state in our methods section: 

“We followed a qualitative descriptive approach to analysis. Our ontological position was relativism, and our epistemological assumptions were grounded in subjectivism. In following this positioning, we viewed this research as both inductive and subjective. We remained aware of the active role of the researcher in co-creating data as it was generated, and also in influencing interpretation through the process of analysis.”

Additionally, in response to this comment we have added further detail about the researchers themselves, and their qualifications (in the methods section), and also explicated what the participants knew about the researchers, to acknowledge the latent power dynamics at play.

“The female trial manager (NMA) with PhD in Biomedical Sciences (Nutrition) and experience of working in clinical trials related to weight management and a female research assistant (RN), who had master’s degree in Public Health Nutrition and prior experience of behavioural interventions conducted the semi-structured interviews with participants.”

We also add transparency in the methods section details about the power dynamics in the analysis process. We state that “ CA kept a reflexivity log, recording personal beliefs, assumptions, and experiences in order to be cognizant of these during analysis, and raising these with NMA during peer-discussions.”

We address the point about the use of the term ‘sub-study’ in our comment above (reviewer 2, comment 1), and have removed this term from our paper.

4. Sampling two - In the Analytic Approach section; the claim that “the end goal was to understand how the acceptability of the programme and the potential for it to be offered at scale” is troubling. For this to be achieved, a sampling strategy that ensured this research represented a broader population, or the population that TDR is aimed at would have to be deployed. As things stand, the knowledge produced in this paper only speaks to the experience of middle-aged white people (we can also make reasonable assumptions about the socio-economic groups that are likely to be represented in the DROPLET trial). This is not to say that these views are not important, but this research is limited in how it can inform any ‘scaling up’.

We agree that this wording may have been misleading, and have therefore removed the refer to how this study can inform scaling-up.

5. The use of the term ‘recruitment bias’ on line 596 further suggests realist assumptions are at the base of this work and raises further questions over the suitability of sampling.

In response to both this comment, and comment 2, we have stated our assumptions explicitly in the methods section. We are confident the sampling is suitable for our stated aims and assumptions, as the sampling is representative of the population who took part in the DROPLET study.

6. Interviews; if a semi-structured interview guide has been based on the experience of the team it would be useful to understand what this is. The knowledge produced in this research is dependent on the researchers interpretative practices. The researchers experience cannot be separated from these practices and good qualitative research would say something about the researcher as much as interviewees. Hiding the researcher, and/or the researchers experience away is cognizant with the objectivity strived for in research that has realist assumptions.

In response to this comment and to reviewer 1, comment 8, we have now included additional information on the interview guide: 

“We developed a semi-structured interview topic guide based on the experience of the team, and the wider qualitative literature on weight management (Supplementary Material). The interview guide was piloted with collages prior to its use with participants. The schedule consisted of twelve open-ended questions with prompts to explore participants’ experience of participating in the DROPLET trial and in particular, views on the TDR programme and type of weight-loss support”

In addition, we’ve included a copy of this guide as supplementary material to the submission.

We are aware these data are constructed and had not intended to hide wither the research or their experience. In response to both this point (and Reviewer 1, comment 3, and reviewer 2, comment 3) we now state clearly both the credentials and the gender of both interviewers, and the qualitative researcher. 

7. Quote; I is not necessary to write ‘um’ in quotations. It means quotes do not read well and does not necessarily change the meaning when removed. As long as the meaning is maintained, I would recommend deleting the ums.

We have removed the ums in the quotes.

8. Reasons for taking part; there is no references in this section, which seem odd. There is a lot of literature that would support these findings; 1). Weight-loss being prompted by wanting to look a different way, 2). Weight-loss being a lifelong thing, 3). The importance of the doctor in getting people to a programme. References to support these findings, and to demonstrate that these findings are not new or unique should be considered.

We have now added references which support the findings in relation to reasons for taking part, and barriers to engagement in the discussion section.

9. The quotes in this section are long considering the points they are supporting are descriptive in nature. Given the suggestion to add text elsewhere, text can be saved here.

We understand this comment. However, our quotes were chosen carefully and we feel they are necessary to illustrate and exemplify our points in the paper. Furthermore, the qualitative descriptive approach requires researchers to remain as close as possible to the participants’ own words, and presenting full, rather than abbreviated quotes, is in line with this approach. 

10. Support and guidance from the counsellor and time to build relationships; again, these findings are interesting but not uncommon. Reference to literature that talks of the importance of the counsellor should be added. These findings are largely descriptive, and while interesting, offer little insight into the importance of the counsellor. An analysis of the importance of empathy and embodiment would provide greater insight into why this programme worked for these 12 interviewees (and why it might not for others). It would be useful to clarify if the counsellor is a paid member of staff or a volunteer, as well as the lay counsellor recruitment (was it necessary they had experience of losing weight?) and their training (were they trained on being empathetic, dealing with a variety of groups etc). The counsellor is obviously an important part of this process, as the literature would suggest, so understanding a bit more about them would strengthen an understanding of why DROPLET worked for a group of white middle-aged adults. Some of this comes in the discussion but would be useful prior to presenting the findings.

Further details on the counsellors has now been provided in the introduction, including their recruitment and training.

We agree an analysis of empathy and embodiment would provide interesting insights in to the role of the counsellor in supporting patients following a TDR programme. However as we state in our title, this is a qualitative descriptive study, we aimed to describe participant experiences across the programme, and analysis of the importance of empathy and embodiment would be the answer to a different research question than the one we address here. We do not have access to data that would support an appropriately rigorous analysis of the importance of empathy and embodiment, but agree this would be a useful area for future study, and have added mention of this to our discussion. We have also added more detail about qualitative descriptive analysis to our section on analytic approach, to highlight that this is a rigorous approach to qualitative enquiry, and is appropriate for studies such as ours which seek to learn about a previously unknown phenomenon, but does not seek to generate theory:

“In qualitative descriptive analysis, the researcher seeks to identify and learn about how people experience an event, or a process, or to learn about people’s perspectives, rather than to generate theory. This method offers opportunity to learn about and to describe phenomena about which little is known. This was appropriate for our research as we do not know about participant experiences of this TDR programme delivered within a routine care context.”

11. These sections effectively highlight the importance of the counsellor but do nothing to critique the bases for this relationship in the current context. Given that there are aspirations to scale up the DROPLET programme, reporting these findings uncritically has the potential to do more harm than good. The questions that remain unanswered here are; what is the context of these individuals that enable them to receive these relationships positively, how the sampling of this study precede these findings, what are the cultural factors at play, how are the lives of these interviewees structured, what place does food play, how does ideology and discourse feature. Downstream programmes that draw on individual responsibility have been shown to widen inequalities. If research like this remains uncritical it only exacerbates and perpetuates this widening and implicates researchers as part of the problem rather than the solution.

We agree that the reviewer raises interesting points here. However, these questions are beyond the scope of this particular study which aimed to describe these participants’ experiences of the TDR itself. We view this paper, which maps experience of a TDR, as a starting point to highlight previously unknown information (such as the importance of the counsellor, and that the diets was reported to be easy and achievable for our sample). A strength of qualitative descriptive studies such as ours is generating this new information about a previously unknown experience, and making it available for other researchers to use as a point to explore some areas in more detail, using different methods, such as something more theoretical, as the reviewer suggests. Following this starting point, we agree it would be important for further studies to focus more closely on an individual’s lifeworld (including the role of food, relationships, family structure, and cultural factors), and if and how aspects of participants lifeworld influenced their ability to engage positively with the TDR. We have added to our discussion that these areas are important to explore future, particularly, as the reviewer states, as there are plans for scale-up and our study (and the DROPLET trial) comprise a relatively homogenous sample.

12. Following the TDR programme; some of the above point are briefly explored here. On first reading I assumed there was a temporal reference in this theme, but on reflection understood it as adhering to the programme. Maybe consider changing this to avoid others making the same mistake.

Thank you for this comment, and we have made this suggested change and have titled the section ‘Adhering to the TDR programme’.

13. Themes adverse effects and recommending TDR to others; it’s hard to see what these themes add. They are descriptive and overly simplistic.

We agree these themes are descriptive and this was in line with our aim to write a descriptive qualitative study of participant experiences of a low-energy total diet replacement programme. We took a descriptive approach here as very little is known about participant TDR experiences within routine care context. In taking a descriptive approach we were able to map breadth of experience across our sample, and throughout the timeline of the diet. This is important for clinicians to know (who may recommend this programme in future) and for other researchers who may be planning TDR studies and can learn what components were described as important by participants. Although these themes may seem simplistic we still think they are important - knowing about the potential of adverse events, what these are and how they can be handled is important learning for researchers, clinicians and patients. We have added further description of the qualitative descriptive approach to our methods section, and outlined our reasoning for selecting this approach.

14. Acceptability; I have issues with the use of this term. This paper effectively shows that 12 white middle-aged interviewees had a positive experience and outcome on a TDR programme. It does not say much, if anything about how and why it is acceptable to this group, or indeed broader social groups (which is necessary for scaling up). An understanding of acceptably can only come by considering the broader social and cultural context. Whether you draw on practice approaches in sociology, or dual-processing theories from psychology, we know that a large part of human action is automatic, unconscious, and beyond rational and cognitive reasoning. I fail to therefore see how the authors can claim that acceptability can be determined from this research. The analysis in this paper is too simplistic to and descriptive to make these claims. The fact the authors have used ‘will go some way’ in line 506 suggests they know or feel that they have made a leap from talking about the descriptive experience of a narrow group of interviewees to talking about acceptability. However, at it’s core I agree with the authors intentions, and qualitative research has an important role to play in the evaluation of these programmes.

We agree that this paper shows that 12 white middle-aged interviewees had a positive experience and outcome on a TDR programme. This is important learning, as experience of a TDR programme delivered within a routine care context was previously unknown. 

Qualitative descriptive studies “are those that represent the characteristics of qualitative research rather than focusing on culture as does ethnography, the lived experience as in phenomenology or the building of theory as with grounded theory”, therefore examination of culture was outside the scope of our data and aims for this study. Our intention was to show how acceptable this diet was to our sample, and have added clarification in our methods section, stating :

“The end goal was to understand the acceptability of the programme for participants in the DROPLET trial.”

15. Discussion; large parts of the discussion provide a similar level of description to the findings, and I therefore feel as though I am reading the same thing again. However, this time there is literature that was missing in the findings.

We endeavoured to keep the results and discussion separate as this is the established format in our field of qualitative applied health services research, and will be most recognizable (and accessible) to our intended audience of other applied health services researchers and clinicians. In our findings section, we aimed to present our findings, remaining close to our data, and in the discussion we ‘step back’ and consider how our findings from this study fit with, differ from, or highlight avenues for future research in this area.

16. One of the stronger parts of this paper is the section on addressing the limitations. However, it reads as if it was written by somebody else as an afterthought. Some of the points here are critical.

Here again we followed the standard format in our field, which includes presenting and discussing ‘strengths and limitations’ in the discussion section. Following this suggestion (and others) we have now additionally presented some of these points elsewhere in the paper, including explicating our sample, and highlighting how the researcher presented themselves and their role in an aim to mitigate social desirability bias.

17. Overall comments; the most compelling part of this paper is the content about lay-counsellors. I would recommend focusing more on this topic by moving beyond description and providing some analysis of these relationship that critique the context they are embedded within. It is also necessary that the authors provide a more honest account of their sampling methods and the claims that are made about what this knowledge can do, based on the obvious limitations in sampling. It would then be appropriate to make claims about acceptability (for a very specific group). It is also necessary for this research to set out the assumptions it is rooted in and consider how there is a level of internal consistency throughout.

We agree the content about lay counsellors is a compelling result. However, this paper maps breadth of reported-experience across the trial. Focussing on the counsellor relationship only would mean other important information about taking part in the trial would be missing

We aimed to map and learn about experiences here, rather than to develop or critique theory, and we are explicit in our title (and throughout) that this is a ‘qualitative descriptive study’. Therefore, we have not moved beyond description to critique context or relationships. Often qualitative descriptive analyses generates key information, which can be used in subsequent, more interpretative studies, and we feel that is what we have done here. In learning about the pivotal role of the councillor, for example, future studies can focus in more depth on this specific aspect of our findings, building upon them, and generative new learning about this phenomena.

We did not set out to be dishonest about our sampling method, and think there may be a misunderstanding here, between our ‘best practice’ on applied health services research to put limitations at the end on the ‘strengths and limitations’ section of the discussion and the reviewer’s research tradition where limitations are written in the methods section itself. Our intent was not to place this information ‘as an afterthought’ or to not be ‘dishonest’ but rather to follow usual practice in our field.

---

## [Editor Report · Decision Letter 1]

21 Aug 2020

Participant experiences of a low-energy total diet replacement programme:  a descriptive qualitative study

PONE-D-20-05095R1

Dear Dr Astbury,

We’re pleased to inform you that your manuscript has been judged scientifically suitable for publication and will be formally accepted for publication once it meets all outstanding technical requirements.

Kind regards,

Louisa Ells, Ph.D.

Academic Editor

PLOS ONE

---

## [Editor Report · Acceptance letter]

28 Aug 2020

PONE-D-20-05095R1 

Participant experiences of a low-energy total diet replacement programme:  a descriptive qualitative study 

Dear Dr. Astbury:

I'm pleased to inform you that your manuscript has been deemed suitable for publication in PLOS ONE. Congratulations! Your manuscript is now with our production department. 

Kind regards, 

on behalf of

Prof Louisa Ells 

Academic Editor

PLOS ONE